# All-Trans Retinoic Acid Exhibits Antiviral Effect against SARS-CoV-2 by Inhibiting 3CLpro Activity

**DOI:** 10.3390/v13081669

**Published:** 2021-08-23

**Authors:** Takeshi Morita, Kei Miyakawa, Sundararaj Stanleyraj Jeremiah, Yutaro Yamaoka, Mitsuru Sada, Tomoko Kuniyoshi, Jinwei Yang, Hirokazu Kimura, Akihide Ryo

**Affiliations:** 1Department of Microbiology, Yokohama City University Graduate School of Medicine, Yokohama 236-0004, Japan; t186075b@yokohama-cu.ac.jp (T.M.); keim@yokohama-cu.ac.jp (K.M.); rediffjerry@gmail.com (S.S.J.); yamaoka-yutaro@kanto.co.jp (Y.Y.); 2Life Science Laboratory, Technology and Development Division, Kanto Chemical Co., Inc., Isehara 259-1146, Japan; 3Advanced Medical Science Research Center, Gunma Paz University, Shibukawa 377-0008, Japan; rainbow_orchestra716@yahoo.co.jp; 4R&D Department, TOKIWA Phytochemical Co., Ltd., Sakura, Chiba 285-0801, Japan; t-kuniyoshi@tokiwaph.co.jp (T.K.); k-you@tokiwaph.co.jp (J.Y.); 5Department of Health Science, Gunma Paz University Graduate School, Takasaki 370-0006, Japan; h-kimura@paz.ac.jp

**Keywords:** 3CL protease, high throughput screening, all-trans retinoic acid, FDA-approved drug, antiviral efficacy, SARS-CoV-2

## Abstract

The pandemic of COVID-19 caused by SARS-CoV-2 continues to spread despite the global efforts taken to control it. The 3C-like protease (3CLpro), the major protease of SARS-CoV-2, is one of the most interesting targets for antiviral drug development because it is highly conserved among SARS-CoVs and plays an important role in viral replication. Herein, we developed high throughput screening for SARS-CoV-2 3CLpro inhibitor based on AlphaScreen. We screened 91 natural product compounds and found that all-trans retinoic acid (ATRA), an FDA-approved drug, inhibited 3CLpro activity. The 3CLpro inhibitory effect of ATRA was confirmed in vitro by both immunoblotting and AlphaScreen with a 50% inhibition concentration (IC_50_) of 24.7 ± 1.65 µM. ATRA inhibited the replication of SARS-CoV-2 in VeroE6/TMPRSS2 and Calu-3 cells, with IC_50_ = 2.69 ± 0.09 µM in the former and 0.82 ± 0.01 µM in the latter. Further, we showed the anti-SARS-CoV-2 effect of ATRA on the currently circulating variants of concern (VOC); alpha, beta, gamma, and delta. These results suggest that ATRA may be considered as a potential therapeutic agent against SARS-CoV-2.

## 1. Introduction

The currently ongoing pandemic of coronavirus disease (COVID-19) caused by severe acute respiratory syndrome coronavirus 2 (SARS-CoV-2) is a global health threat [1]. Recent efforts in vaccination offer a ray of hope in providing widespread protection, but the rise in antibody escape mutants complicates the scenario [2]. Using efficient antivirals to treat the infected can reduce the viral loads, thereby decreasing the possibility of transmission and evolution of vaccine escape mutants. However, there are only a few antiviral agents that are currently available for treating this disease [3].

The SARS-CoV-2 has two viral proteases, namely the main protease also called 3CL protease (3CLpro), and the papain-like protease of which the former plays an important role in viral replication. 3CLpro has high specificity to a unique amino acid recognition sequence on its substrate which is not shared by any of the known human proteases [4]. Because of this specificity, 3CLpro is one of the most interesting targets for antiviral drug development [4,5]. Several 3CLpro inhibitors have been reported for the earlier coronaviruses [6] and also the ones such as N3 [7], 13b [4], and GC376 [8] have been reported for SARS-CoV-2. However, none of these drugs are approved for human use yet.

Vitamin A (VA) comprises the group of fat-soluble retinoids including retinol and retinal. Retinoic acid (RA), the bioactive form of retinol, exists as three isomers: all-trans retinoic acid (ATRA), 9-cis retinoic acid (9cis-RA), and 13-cis retinoic acid (13cis-RA). ATRA, also known as Tretinoin, is a food and drug administration (FDA) approved drug for the treatment of acute promyelocyte leukemia (APL). RA is known to play complex roles in the immune system. Type I interferons (IFN-I) are one of the most powerful antiviral mediators of host defense, and RA mediates their synthesis through nuclear retinoic acid receptors RAR and RXR [9]. Previous studies have demonstrated that ATRA has antiviral activity against several viruses [10,11,12]; principally by enhancing the IFN-I mediated viral clearance. Apart from this, ATRA has also been demonstrated to possess IFN-I independent antiviral effects [13,14].

In this study, we developed a quantitative high throughput screen system that can evaluate the inhibitory activity for SARS-CoV-2 3CLpro using AlphaScreen, which provides quantitative detection of 3CLpro activity with a wider dynamic range and higher sensitivity than fluorescence resonance energy transfer (FRET) [15]. Using this system, we screened a library of natural compounds and found that ATRA efficiently inhibits the 3CLpro activity of SARS-CoV-2. Furthermore, we investigated the antiviral activity of ATRA against SARS-CoV-2 in cell culture and showed that ATRA has a direct antiviral effect by inhibiting 3CLpro of SARS-CoV-2.

## 2. Materials and Methods

### 2.1. Cells, Virus and Compounds

VeroE6/TMPRSS2 cells (JCRB #1819) [16] and Calu-3 cells were cultured in DMEM containing 10% FBS. SARS-CoV-2 Pango lineage A (JPN/TY-WK-521/2020 EPI_ISL_408667), alpha strain (JPN/QK002/2020 EPI_ISL_768526), beta strain (JPN/ TY8-612-P1/2021 EPI_ISL_1123289), gamma strain (JPN/TY7-501/2021 EPI_ISL_833366), and delta strain (JPN/TY11-927-P1/2021 EPI_ISL_2158617) were obtained from NIID, JAPAN, and handled in biosafety level 3 (BSL3). While performing SARS-CoV-2 infection experiments, DMEM containing 2% FBS was used. The chemical compound library was obtained from Tokiwa Phytochemical (Chiba, Japan) and retinoids including ATRA, all-trans retinal, retinol, 9-cis RA, and 13-cis RA were obtained from Sigma Aldrich (St. Louis, MO, USA). GC376 was purchased from Selleck chemicals (Houston, TX, USA).

### 2.2. Wheat Germ Cell-Free Protein Synthesis and Protein Purification

In vitro wheat germ cell-free protein synthesis was carried out as previously described [17,18,19]. Protein synthesis of FLAG-SARS-CoV-2-nsp4/5-GST-biotin was performed using WEPRO7240G wheat germ extract (CellFree Sciences, Yokohama, Japan) in the bilayer translation reaction. Synthesized proteins were confirmed by SDS-PAGE followed by CBB staining with RapidCBB KANTO 3S (Kanto chemical, Tokyo, Japan) and immunoblotting. FLAG-SARS-CoV-2-nsp4/5-GST-biotin was purified by Glutathione Sepharose 4 Fast Flow (Cytiva, Waukesha, WI, USA).

### 2.3. SARS-CoV-2 3CLpro Activity Assay

Protease activity assay, 100 nM of recombinant SARS-CoV-2 3CL protease (#E-718-050, R&D SYSTEMS) was mixed with compounds and pre-incubated at room temperature for 30 min. After incubation, FLAG-nsp4/5-GST-biotin was mixed with protease and compounds mixture and incubated for 4 h at room temperature. Detection of protease activity was performed essentially as described in the AlphaScreen FLAG (M2) Detection Kit instruction manual (PerkinElmer, Boston, MA, USA). 10 μL of detection mixture containing 20 mM Tris-HCl (pH 7.5), 0.2 mM DTT, 5 mM MgCl2, 1 mg/mL BSA, 0.1 μL streptavidin-coated donor beads, and 0.1 μL anti-FLAG acceptor beads were added to each well of the 384-well plate, followed by incubation at 26 °C for 1 h in the dark. Luminescence signals were analyzed with the AlphaScreen detection program using the EnSpire software (PerkinElmer).

The cleavage of substrate proteins was analyzed by immunoblotting using HRP-conjugated streptavidin (1:10,000, #RPN1231v, GE Healthcare Bioscience, Piscataway, NJ, USA). Inhibition of substrate cleavage by GC376 (final concentration was 50 µM) and ATRA (final concentration was 0.1 µM–100 µM) was performed by pre-incubation with the protease for 30 min at room temperature, followed by reaction with the substrate.

FRET-based protease activity assay was performed using 3CL Protease, Untagged (SARS-CoV-2) Assay Kit (BPS Bioscience, CA, USA) according to manufacturer’s instructions. The fluorescence intensity was measured by SparkControl (Tecan, Männedorf, Switzerland), at excitation wavelength 360 nm and detection of emission at a wavelength 460 nm.

### 2.4. Docking Simulation

The X-ray crystallographic structure data of SARS-CoV-2 3CLpro (PDB: 6YB7) was retrieved from the RCSB website. The PDB data of ligand, tretinoin (ATRA) was retrieved from DrugBank. The calculations for the molecular docking simulation were performed using AutoDock Vina [20] and the results were analyzed using PyMOL.

### 2.5. Infectivity Assay

VeroE6/TMPRSS2 and Calu-3 cells were pre-treated with compound for 3 h at 37 °C, were infected with SARS-CoV-2 (MOI = 0.05 for VeroE6/TMPRSS2 cells and MOI = 0.5 for Calu-3 cells) for 2 h. Then, the virus-drug mixture was removed, and cells were cultured with fresh drug and 5% FBS containing medium. Cell viability assay and RT-PCR were performed 48 h (VeroE6/TMPRSS2 cells) and 72 h (Calu-3 cells) after infection. All the infection experiments were performed at BSL-3.

### 2.6. Cell Viability Assay

Cell viability was measured using CellTiter-Glo (Promega Corporation, Madison, WI, USA) which quantitatively detects live cells based on ATP levels. 40 µL of CellTiter-Glo substrate was added to the cells and their viability was measured based on the luminescence detected by GloMax Discover System (Promega Corporation) after 10 min. 

### 2.7. RNA Extraction and Real-Time Reverse Transcriptase Quantitative Polymerase Chain Reaction (RT-qPCR)

Virus RNA was extracted at supernatants by QIAamp Viral RNA Mini Kit (Qiagen, Hilden, Germany) according to the manufacturer’s instructions. The viral RNA was quantified using CFX96 Real-Time System (Bio-Rad, Hercules, CA, USA) with a TaqMan Fast Virus 1-Step Master Mix (Thermo Fisher Scientific, Waltham, MA, USA)) using 5′-AAATTTTGGGGACCAGGAAC-3′ and 5′-TGGCAGCTGTGTAGGTCAA-3′ as the primer set and 5′-FAM-ATGTCGCGCATTGGCATGGA-BHQ-3′ as a probe. 

### 2.8. Immunofluorescence Microscopy

Cells were fixed with 4% paraformaldehyde and permeabilized with 0.5% Triton X-100, and then were blocked with Blocking One (Nakalai Tesque, Kyoto, Japan) at room temperature for 15 min. The cells were incubated with the polyclonal antibody against SARS-CoV-2 Nucleocapsid Protein antibody (1:100 dilution, #NB100-56576, Novus) at room temperature for 1 h. After incubation, cells were stained with Alexa 568-labeled anti-rabbit antibody (1:1000 dilution, Thermo Fisher Scientific, Waltham, MA, USA) for 1 h at room temperature. The nucleus was stained with ProLong Gold Antifade Mountant with DAPI (Thermo Fisher Scientific). The images were taken and measured by fluorescence microscopy, BZ-X810 (Keyence, Osaka, Japan).

### 2.9. Time of Addition Assay

The time of addition assay was performed at three different times: “Full-time”, “Entry”, and “Post-entry”. In “Full-time” treatment, VeroE6/TMPRSS2 cells pre-treated with compounds (25 µM of ATRA, 10 µM of remdesivir, and 25 µM of camostat) for 3 h were infected with virus 2 h after which, the medium was removed, and replaced with compounds containing medium and incubated until the end of the experiment at 48 h. For “Entry” treatment, cells were pretreated similarly for 3 h prior to infection, followed by 2 h of virus exposure, after which the medium was replaced with a fresh culture medium and incubated for 48 h. For “Post-entry” treatment, cells were first infected with the virus, and compounds were added at 2 h post-infection and maintained until the end of the experiment at 48 h. Time of addition assay was evaluated by calculating the copy number of SARS-CoV-2 in the culture medium 48 h after infection by RT-PCR.

## 3. Results

### 3.1. Development of the SARS-CoV-2 3CLpro Enzyme Assay Using AlphaScreen and Screening of Compounds That Inhibit Enzyme Activity

To develop the quantitative high-throughput screening of SARS-CoV-2 3CLpro inhibitors based on AlphaScreen, we synthesized using wheat germ cell-free system, a recombinant protein substrate (FLAG-TSITSAVLQ^SGFRKMAFP-GST-biotin) that is cleaved by SARS-CoV-2 3CLpro. This recombinant substrate could be bound by an anti-FLAG antibody fused with acceptor beads at one end and streptavidin fused with donor beads at the other end. Intact recombinant substrate emits a high luminescence signal due to the close proximity between the acceptor and donor beads, and if it is cleaved by 3CLPro, the donor and acceptor beads drift apart to emit a low signal (Figure 1A). To confirm whether the cleavage of the recombinant substrate could emit detectable signals, we mixed wild type (WT) 3CLpro or its inactive mutant C145A (active site Cys^145^ is replaced by Ala), with the substrate and analyzed the emitted signals. WT 3CLpro reduced the signal, while C145A did not. Furthermore, immunoblots revealed a substrate cleavage band with WT and not with C145A (Figure 1B). To optimize the assay condition of the system, different concentrations of 3CLpro were allowed to react with 100 nM of the recombinant substrate in 384-well plates and the intensity of the signal was measured (Figure 1C). The SARS-CoV-2 3CLpro inhibition test was performed using GC376, which has been shown to inhibit 3CLpro activity. GC376 was incubated with 3CLpro for 30 min at room temperature, and then the recombinant substrate was added and incubated for 4 h at 37 °C. GC376 showed inhibitory activity against 3CLpro in a concentration-dependent manner (Figure 1D). Using this system, we screened 91 naturally occurring compounds at 50 µM each (Appendix A) for their ability to inhibit SARS-CoV-2 3CLpro. Our results showed that ATRA was the only compound in the panel to inhibit the activity of 3CLpro by more than 50% (Figure 1E). 

To assess the performance of our newly developed AlphaScreen assay, we compared it with the conventional fluorescence resonance energy transfer (FRET) assay. The AlphaScreen assay had a better signal-to-noise ratio (S/N) than the FRET indicating the higher sensitivity of the former (Appendix A). We then ran the FRET assay on the same sample set of 91 compounds and found that FRET identified only four of the seven compounds detected by the AlphaScreen assay and ATRA showed more than 50% inhibitory activity in both the assays (Appendix A). We thus selected ATRA for further functional analysis.

### 3.2. ATRA as a Potent SARS-CoV-2 3CLpro Inhibitor 

Since ATRA showed a prominent effect in inhibiting SARS-CoV-2 3CLpro, we hypothesized that other retinoids might also have a similar effect. Hence, we performed the SARS-CoV-2 3CLpro activity assay using ATRA and four of its analogs; all-trans-retinal, retinol, 9cis-RA, and 13cis-RA (Figure 2A). These results showed that ATRA had a potent SARS-CoV-2 3CLpro inhibition effect compared to other retinoids and was hence chosen for further analysis (Figure 2B). A concentration gradient of ATRA was prepared, mixed with 3CLpro, and then reacted with the recombinant substrate to show the inhibitory activity of ATRA on 3CLpro by immunoblot and AlphaScreen. In immunoblots, cleavage bands of the substrate appeared at ATRA concentrations of 10 µM and below (Figure 2C). In AlphaScreen, the signal decreased in a concentration-dependent manner and the 50% inhibitory concentration of the signal, IC_50_ was 24.7 ± 1.65 µM (Figure 2D).

In order to better understand the basis for its inhibition, we created a docking model using AutoDock Vina to analyze the interaction of ATRA with 3CLpro [20]. Docking simulation predicted that ATRA interacts with the active site of SARS-CoV-2 3CLpro (PDB: 6YB7). SARS-CoV-2 3CLpro has the Cys-His catalytic site containing active residues of Cys^145^ and His^41^ located in the substrate-binding domain (residues 163–167 and 178–192). ATRA was seen to fit the substrate-binding pocket (Figure 2E), with the formation of two H-bound interactions with Thr^190^ with the binding distances of 1.9 Å and 3.1 Å (Figure 2F) at a docking score of −5.9 kcal/mol with root mean square deviation (RMSD) of 1.2 Å.

In order to further evaluate these predictions, we obtained Lineweaver-Burk plots to identify the mode of inhibition for several compounds [21]. These studies used a fixed 100 nM concentration of 3CLpro and increasing substrate and ATRA concentrations. In contrast with the expectations, the results indicate that the effect of ATRA on 3CLpro activity primarily follows a non-competitive mode of inhibition (Appendix A). Thus, although the inhibitory effect of ATRA may to some extent result from active site binding, it results primarily from binding to an additional site or sites. 

### 3.3. ATRA Inhibits SARS-CoV-2 Replication

Since previous studies have shown that some compounds with SARS-CoV-2 3CLpro inhibitory activity do not inhibit SARS-CoV-2 replication [22], we investigated whether ATRA possessed an antiviral effect on SARS-CoV-2. Further, ATRA is known to synergistically enhance IFN-I production in virus-infected cells [11] by upregulating RIG-I and IRF1 [23]. To avoid the confounding effect of IFN-I on the anti-SARS-CoV-2 3CLPro action of ATRA, we used VeroE6/TMPRSS2 cells, which are not capable of producing IFN-I [24]. VeroE6/TMPRSS2 cells were treated with ATRA and then infected with SARS-CoV-2, and inhibition of cell death was evaluated by cell viability assay. RT-PCR assay was performed to quantify SARS-CoV-2 RNA in the culture supernatant (Figure 3A). The results showed that ATRA had the antiviral effect against SARS-CoV-2 with IC_50_ = 8.19 ± 0.06 μM for cell viability assay (Figure 3B) and IC_50_ = 2.69 ± 0.09 µM for RT-PCR assay (Figure 3C). Additionally, ATRA did not exhibit any cytotoxicity at high concentrations; a concentration of cytotoxicity (CC_50_) > 100 µM (Appendix A).

In the immunofluorescence analysis, we confirmed infection in cells by visualizing SARS-CoV-2 nucleocapsid protein (NP) expression through immunofluorescence microscopy at 24 h post-infection (cytopathic effects were not obvious at this time-point of infection). We compared the efficacy of ATRA (1, 10, 25 µM) against 10 µM of remdesivir [25]. ATRA showed a concentration-dependent decrease in the number of infected cells (Figure 3D). To identify the part of the viral life cycle blocked by ATRA, we utilized a time-of-drug addition approach. We tested the antiviral activity of ATRA added at three different time points (Figure 3E). Remdesivir, which shows anti-SARS-CoV-2 effects at the post-entry stage [25], and camostat at the entry stage [26], were used for comparison. The time of addition assay showed that ATRA inhibited SARS-CoV-2 replication in VeroE6/TMPRSS2 cells at the post-entry stage similar to remdesivir (Figure 3F,G), while camostat functioned at entry stages (Figure 3H). These data indicate that ATRA acts intracellularly to inhibit viral replication after entry, consistent with an inhibitory effect of ATRA on 3CLpro.

### 3.4. ATRA Inhibits SARS-CoV-2 Replication Independent of RIG-I Expression

Infection experiments using VeroE6/TMPRSS2 cells showed that ATRA had anti-SARS-CoV-2 effects in an IFN-I-independent manner. Furthermore, RIG-I has a direct antiviral effect on SARS-CoV-2 [27]. To avoid the effect of ATRA on RIG-I-mediated anti-viral action, we used Calu-3 cells derived from human lung epithelium since this cell line has been reported to have very low expression of RIG-I [27]. We performed RT-PCR assay on the supernatants of infected cells treated with 1–100 µM of ATRA (Figure 4A) revealed that ATRA inhibited the replication of SARS-CoV-2 in a dose-dependent manner, showing IC_50_ = 0.82 ± 0.01 µM and the concentration at which the virus copy number was reduced by 3-log was 11.5 ± 0.01 µM (Figure 4B). There was no observable cytotoxicity even at a concentration of 100 µM (Appendix A). In addition, we did not find any upregulation of RIG-I by ATRA in infected cells (Appendix A). These results indicate that ATRA potently inhibits the replication of SARS-CoV-2 in human lung cells, independent of the induction of RIG-I expression.

### 3.5. ATRA Is Effective against the SARS-CoV-2 Variants of Concern in Human Lung Cell Line

Currently, there are different variants of concern (VOC) of SARS-CoV-2 which have mutated from the first isolated Pango lineage A and pose problems with respect to transmissibility, disease severity, evasion from neutralization, and failure of treatments and vaccines [28]. We investigated the anti-SARS-CoV-2 effect of ATRA on the common VOC in circulation, alpha, beta, gamma, and delta. Bioinformatic alignment using Clustal Omega revealed that the alpha, gamma, and delta strains did not possess any mutations in the entire 306 amino acid sequence of 3CLpro, while the beta strain contained a single point mutation K90R. The active site and substrate binding site of 3CLpro were conserved among all the VOC suggesting that ATRA would be effective on these strains (Appendix A). To confirm this, we examined the anti-SARS-CoV-2 effect of ATRA on the VOC strains using Calu-3 cells by RT-PCR assay. RT-PCR results showed that ATRA suppressed replication of all the VOC strains with the efficacy as observed with the original isolate (Figure 4B–F), suggesting that ATRA can be an effective antiviral against the commonly circulating SARS-CoV-2 VOC.

## 4. Discussion

The pandemic of COVID-19 is still spreading worldwide due to the lack of effective antiviral measures. Several studies have demonstrated that SARS-CoV-2 3CLpro could be essential for viral replication and considered to be an important therapeutic target against COVID-19 [4,5]. Although previous studies attempted to screen potent 3CLpro inhibitors using conventional methods such as fluorescence resonance energy transfer (FRET), only a few promising compounds that have been approved for human use have been identified. This could be due to the inherent low sensitivity or high background of the assay platform [15]. To overcome these obstacles, we here developed a novel in vitro assay system using AlphaScreen for 3CLpro capable of high throughput. This system enables us to analyze the enzyme activity of 3CLpro by utilizing a cell-free-synthesized reporter substrate that can be targeted by both anti-FLAG acceptor beads and streptavidin-coated donor beads. The AlphaScreen assay provides several advantages based on its unique features over alternative technologies thereby achieving high sensitivity and low background with optimal versatility in assay design. By utilizing this technology, we screened a natural compound library and identified ATRA as a potent inhibitor against 3CLpro. Furthermore, the compounds that showed only weak inhibitory activity by FRET were also detected by Alphascreen, suggesting that this method used in the present study is useful for a highly sensitive search for compounds that inhibit 3CLpro. Since ATRA has been approved by FDA in the treatment of acute promyelocytic leukemia, its indication to COVID-19 could be desirable [29].

We observed that ATRA inhibited SARS-CoV-2 3CLpro in a non-competitive manner. It is known that non-competitive inhibition is mediated by allosteric sites, which are sites other than the active site [30]. In this study, ATRA was predicted to bind to Thr190, but since this is only a prediction of binding in simulation, further studies are needed to identify the actual binding site.

Previous studies have reported that ATRA upregulates RIG-I in virus-infected cells and exhibits antiviral effects in an IFN-I-dependent manner [11,12]. In addition, Yamada et al. reported that RIG-I directly binds to SARS-CoV-2 RNA and inhibits viral replication, thereby exerting an antiviral effect [27]. Following our in vitro screening, we performed cell culture-based analysis to examine the effect of ATRA on the viral infection through the inhibition of 3CLpro. To avoid the effect of ATRA on RIG-I-mediated innate immune response, we used Vero cells in which the genes for interferon synthesis are defective [24]. We also used Calu-3 cells that exhibit very low expression of RIG-I [27]. Indeed, we observed no increase in RIG-I expression with ATRA treatment both in SARS-CoV-2-infected and un-infected cells. Our current result indicates that SARS-CoV-2 infection was potently suppressed by ATRA in VeroE6/TMPRSS2 and Calu-3 cells showing a RIG-I-independent antiviral effect.

ATRA has been shown to directly affect viral replication by interfering with viral components such as the reverse transcriptase enzyme of human immunodeficiency virus-1 (HIV-1) and human T-cell lymphoma virus-1 (HTLV-1) [13,14]. In our study, we reveal the interaction of ATRA with SARS-CoV-2 3CLPro. Virus-encoded proteases have been shown to be involved in the replication of many RNA viruses [31] and therefore can be a key drug target. Indeed, the major success of anti-retroviral therapy by targeting HIV protease suggests that viral proteases can be valid molecular targets for RNA viruses. We observed the inhibitory effect of ATRA on 3CLpro in vitro at an IC_50_ of 24.7 ± 1.65 µM while 3-log reduction of virus copies in Calu-3 cells was achieved at a lesser concentration of 11.5 ± 0.01 µM. Moreover, anti-viral action of ATRA on virus replication was shown to be much stronger than its inhibitory effect on 3CLpro, demonstrating IC_50_ = 2.6 ± 0.09 µM and 0.82 ± 0.01 µM in VeroE6/TMPRSS2 and Calu-3, respectively. These differences suggest that ATRA has multiple antiviral effects other than inhibiting 3CLpro activity. ATRA is known to exhibit many physiological activities such as cell differentiation and immunity [32,33,34,35]. 

Currently, there are various mutants and VOC of SARS-CoV-2. Vaccines are effective in preventing infections, but it is feared that their effectiveness may be diminished with the emergence of new mutants [28]. This problem could be reduced if an effective antiviral can be developed to reduce the viral reservoir in infected individuals. Remdesivir, favipiravir, camostat mesylate, ivermectin are some of the antivirals which are being used in the treatment of COVID-19 [36]. Additionally, we here reveal that ATRA could be a potential antiviral agent against COVID-19.

ATRA can be administered orally to achieve peak plasma concentrations of 0.1–8 μM [37]. Since we observe that higher concentrations of ATRA are required for SARS-CoV-2 inhibition in vitro, the in vivo antiviral effect of ATRA remains to be studied. HIV protease inhibitors such as lopinavir and ritonavir were expected to be therapeutic agents for SARS-CoV-2 as they showed a good effect in vitro, but proved to be less effective when used for treatment [38]. Since ATRA is already an FDA-approved drug for APL, its human use in COVID-19 could be expedited if the antiviral effect could be confirmed in animal studies.

## Figures and Tables

**Figure 1 viruses-13-01669-f001:**
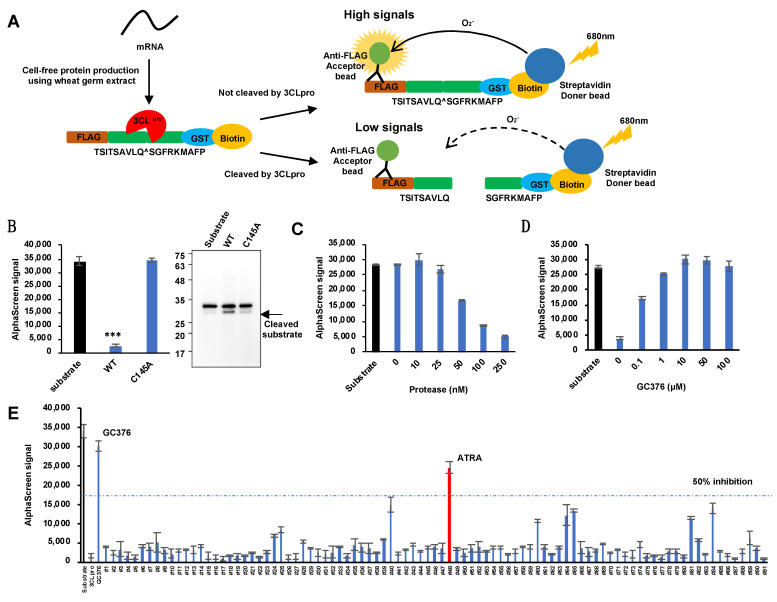
Development of SARS-CoV-2 3CLpro inhibitor screening assay using AlphaScreen. (**A**) Schematic representation of AlphaScreen-based assay for SARS-CoV-2 3CLpro enzymatic activity. A high signal is detected when the substrate is not cleaved by 3CLpro, and the signal becomes weaker when the substrate is cleaved. (**B**) Results of an enzymatic assay using wild-type SARS-CoV-2 3CLpro and its inactive mutant, C145A. 3CLpro decreases the signal when the substrate is cleaved. Substrate cleavage was also detected by immunoblots. Error bars obtained from duplicate testing. *p*-value < 0.005 (***). WT-Wild type. (**C**) Optimization of the enzyme assay. A concentration gradient of 3CLpro was reacted with 100 nM of the recombinant substrate and the signal was detected. Error bars obtained from duplicate testing. (**D**) The inhibitory effect of 3CLpro using the known SARS-CoV-2 3CLpro inhibitor GC376. Error bars obtained from duplicate testing. (**E**) Screening of 91 compounds using the 3CLpro enzyme assay. The substrate and enzyme concentration was 100 nM in this experiment. GC376 was used as a positive control. ATRA showed in the red bar. Error bars obtained from duplicate testing.

**Figure 2 viruses-13-01669-f002:**
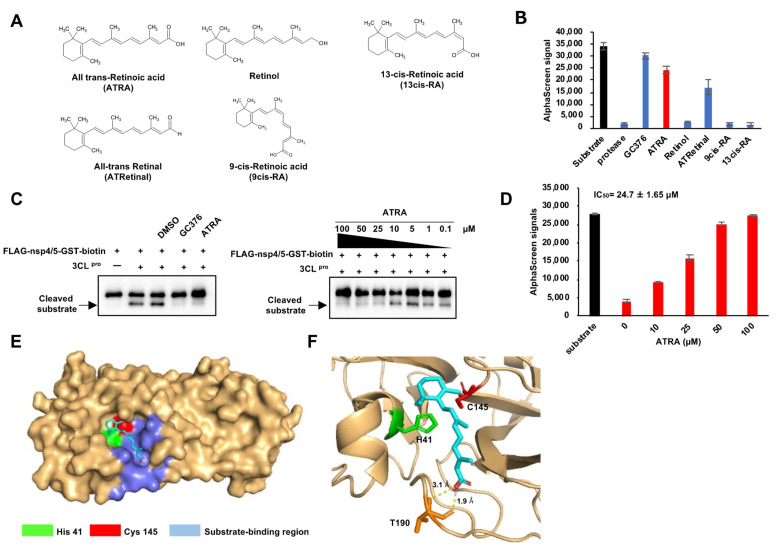
ATRA as a potent SARS-CoV-2 3CLpro inhibitor (**A**) The structure of retinoids used in this study. (**B**) The results of enzyme assay for different retinoids (ATRA; colored red). Error bars obtained from duplicate testing. (**C**) In vitro cleavage assay, the recombinant substrate was cleaved by SARS-CoV-2 3CLpro, which was confirmed by immunoblot. A cleaved substrate formed a band right below the substrate band. Substrate cleavage by SARS-CoV-2 3CLpro was inhibited by GC376 (100 µM) and ATRA (0.1–100 µM). (**D**) Results of the enzyme assay for ATRA in a concentration gradient (0–100 µM, colored red). Error bars obtained from duplicate testing. (**E**,**F**) Docked poses of ATRA and SARS-CoV-2 3CLpro showed as surface (**E**) and ribbon (**F**). The catalytic residues are His ^41^ (colored green) and Cys ^145^ (colored red). The substrate-binding region is 163–167 and 178–192 (colored blue).

**Figure 3 viruses-13-01669-f003:**
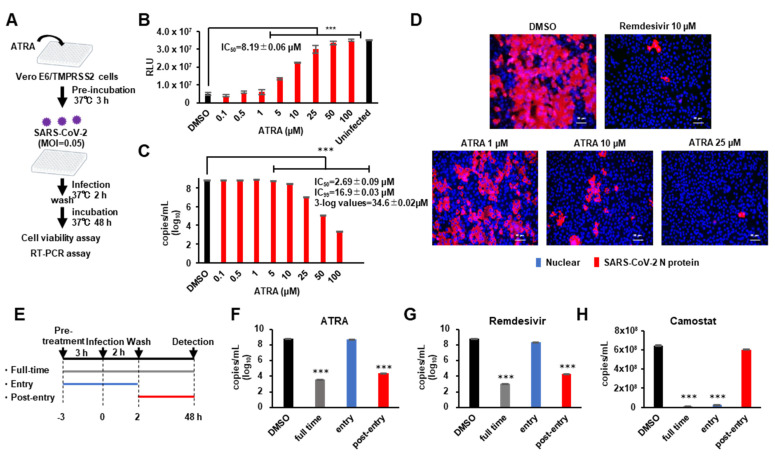
ATRA inhibits SARS-CoV-2 replication (**A**) Assessing the antiviral effect of ATRA in VeroE6/TMPRSS2 cells. VeroE6/TMPRSS2 cells treated with ATRA (0.1–100 µM) for 3 h prior to infection were infected with SARS-CoV-2 (MOI = 0.05) and washed off the unbound virus at 2 h of infection. Cell viability assay and RT-PCR were performed on the attached cells and supernatant culture medium respectively, 48 h after infection. (**B**) The results of the cell viability assay. IC_50_ was calculated for the inhibition of SARS-CoV-2 induced cell death. Error bars obtained from triplicate testing. *p*-value < 0.005 (***) (**C**) Viral quantification by RT-PCR. Error bars obtained from duplicate testing. *p*-value < 0.005 (***) (**D**) Immunostaining of SARS-CoV-2 N protein in infected cells. VeroE6/TMPRSS2 cells treated with ATRA (1, 10, 25 µM) 3 h before infection were infected with SARS-CoV-2 (MOI = 0.05). After 24 h of infection, cells were fixed with 4% paraformaldehyde and immunostained. Red stained cells represent SARS-CoV-2 N protein, blue-stained nuclei with DAPI. (**E**) Schematic representation of the time of addition assay of ATRA. (**F**-**H**) The results of RT-PCR for ATRA (**F**), Remdesivir (**G**) and Camostat (**H**). The result of full-time colored grey, entry colored blue, and post-entry colored red. Error bars obtained from duplicate testing. *p*-value < 0.005 (***).

**Figure 4 viruses-13-01669-f004:**
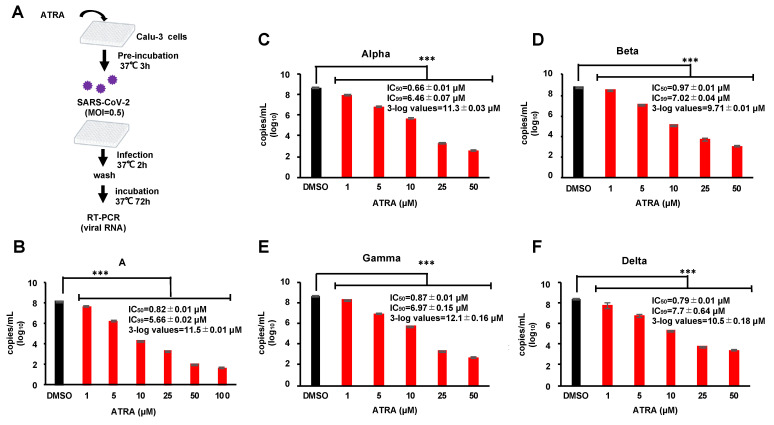
ATRA is effective against the SARS-CoV-2 variants of concern in the human lung cell line (**A**) Calu-3 cells, pre-incubated with ATRA at 37 °C for 3 h, were infected with SARS-CoV-2; A, alpha, beta and gamma (MOI = 0.5) for 2 h. Following this, the non-adsorbed virus was washed, and the infected cells were incubated in a fresh medium containing ATRA for 72 h. (**B**–**F**) Inhibition of different VOC by ATRA. The results of RT-PCR for lineage A (**B**), alpha strain (**C**), beta strain (**D**), gamma strain (**E**), and delta strain (**F**). Error bars obtained from duplicate testing. *p*-value < 0.005 (***).

## Data Availability

Not applicable.

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
