# Peer review of "All-Trans Retinoic Acid Exhibits Antiviral Effect against SARS-CoV-2 by Inhibiting 3CLpro Activity"

_viruses, 2021, doi:10.3390/v13081669_

Round 1

Reviewer 1 Report

In this revised manuscript, the authors have responded fully to all of the suggestions made in the initial review.  Unfortunately, there is a problem with the response to point 2, which is contained in section 3.2, lines 209-222. The authors first present the experimental kinetic results showing non-competitive inhibition, and then present the modeling results predicting competitive results.  What is required here is to present the initial modeling results first, then provide the experimental results that in this case (and contrary to both my and the authors expectations) demonstrated non-competitive inhbition, and then to provide some discussion or insight into the basis for the difference.  In order to clarify what is needed, I have rearranged the paragraph on lines 209-222 below:

Revised paragraph in results section:

In order to better understand the basis for its inhibition, we created a docking model using AutoDock Vina to analyze the interaction of ATRA with 3CLpro [20]. Docking simulation predicted that ATRA interacts with the active site of SARS-CoV-2 3CLpro (PDB: 6YB7). SARS-CoV-2 3CLpro has the Cys-His catalytic site containing active residues of Cys145 and His41 located in substrate binding domain (residues 163-167 and 178-192). ATRA was seen to fit the substrate-binding pocket (Figure 2E), with formation of two H-bond interactions with Thr190 corresponding to binding distances of 1.9 â„« and 3.1 â„« (Figure 2F) at a docking score of -5.9 kcal/mol with root mean square deviation (RMSD) of 1.2 â„«.

In order to further evaluate these predictions, we obtained Lineweaver-Burk plots to identify the mode of inhibition for several compounds [21]. These studies used a fixed 100 nM concentration of 3CLpro and increasing substrate and ATRA concentrations. In contrast with expectations, the results indicate that the effect of ATRA on protease activity primarily follows a non-competitive mode of inhibition (Figure S2). Thus, although the inhibitory effect may to some extent result from active site binding, it results primarily from binding to an additional site or sites.

The authors probably want to further revise this material, including renumbering of the corresponding references. But the experimental results supersede the predictions about where the inhibitor binds. 

Author Response

We highly appreciate the reviewer's comments and valuable suggestion that significantly improve our paper. Accordingly, we have rephrased the sentence in section 3.2 lines 210-225. 

Reviewer 2 Report

The authors present the results of an in vitro study on the antiviral activity of all-trans retinoic acid against SARS-CoV-2 and the analysis of molecular mechanisms associated with the antiviral activity. The study is clearly presented and the results are substantiated by data. The significance of these in vitro results is discussed in a reasonable and objective manner. I recommend the study for publication in the present form. 

Author Response

We are thankful for this positive comment from the reviewer.

This manuscript is a resubmission of an earlier submission. The following is a list of the peer review reports and author responses from that submission.

Round 1

Reviewer 1 Report

This study reports on the development of an improved, higher sensitivity assay to measure the activity of the SARS-CoV-2 main protease (3CLPro), and the application of this approach to the identification of a natural product – all trans retinoic acid (ATRA) that shows significant inhibitory activity.  The study also demonstrates that ATRA inhibits viral replication several different cell lines, helping to more specifically identify the inhibitory mechanism. The presentation is clear and well organized. The urgency of the current pandemic and the value of demonstrating that a drug, previously approved for the treatment of acute promyelocytic leukemia, may be useful against COVID-19 strongly support publication.  There are a few specific areas in which the paper could be improved, however I do not feel that implementing these changes is essential:

  • In general, it is useful to demonstrate the advantage of a new, higher sensitivity assay by a direct comparison of the results obtained on a given sample set using previous (FRET) technology.  The authors in fact suggest that the limited number of inhibitors identified using the FRET assay could be a consequence of its limited sensitivity. Thus it would be useful to directly compare results using both assays on the same sample set, e.g. the sample set used in the present study.  Perhaps the authors can identify specific compounds that show significant activity in their assay but not in the earlier assays?

  • The modeling reported by the authors is quite plausible, but would be more directly supported by a demonstration that the 3CLPro inhibition is competitive rather than non-competitive. There is some evidence for the existence of an allosteric 3CLPro site (Du et al., Antiviral Res., 2021); further, some of the other natural products tested, such as #40 – (-)-epigallocatechin, look a bit similar to the allosteric ligands such as chebularic acid, described by Du et al.

  • In Figure S3, the authors report the alignment of 3CLPro variants of concern. It would be useful to more clearly identify mutations in this Figure, and also to include the sequence for the delta variant (B.1.617.2), even though the authors have not explicitly performed studies on this variant, in order to identify potential active site variations.

Additional minor typos and grammatical points are summarized below:

Line 47 – huma should be human

Line 143 –  replace "condition" with "assay conditions" 

Line 206-207 – change "and possibly hinder its activity" to "consistent with the inhibition data". 

Lines 203, change "binding to Thr190" to "formation of two H-bond interactions with Thr190".

The authors might also mention that the ATRA ring interacts with active site residues Cys145 and His41. 

Line 243 replace "suggesting" with "consistent with"

Line 277 – what is bioinformatic alignment?  Did the authors use clustal omega or another program?

Line320 – replace "prominently" with "potently" or "strongly"

The bar graphs in the figures use different color bars. Although the labeling indicates what these correspond to, it would probably be a good idea to identify what the different colors mean in the captions. 

Reviewer 2 Report

The authors present data on the antiviral effect of all-trans retinoid acid on 3CLpro of SARS-CoV-2, including the variants of concern. The results of are an important scientific contribution to the field.. Study design and methodology are sufficiently described and the results are presented clearly. Conclusions are based on results and discussion is appropriate. I recommend the publication of the paper in the present form.